# RC-SLAM: Road Constrained Stereo Visual SLAM System Based on Graph Optimization

**Yuan Zhu** [1], **Hao An** [1], **Huaide Wang** [1], **Ruidong Xu** [1], **Mingzhi Wu** [2] **and Ke Lu** [1,*]

[1] School of Automotive Studies, Tongji University, Shanghai 201800, China; yuan.zhu@tongji.edu.cn (Y.Z.); hao_an@tongji.edu.cn (H.A.); huaide_wang@tongji.edu.cn (H.W.); rd_xu@tongji.edu.cn (R.X.)
[2] Nanchang Automotive Institute of Intelligence & New Energy, Tongji University, Nanchang 330038, China; wumingzhi@naiine.com
[*] Correspondence: luke@tongji.edu.cn; Tel.: +86-136-6197-8920

**Abstract:** Intelligent vehicles are constrained by road, resulting in a disparity between the assumed six degrees of freedom (DoF) motion within the Visual Simultaneous Localization and Mapping (SLAM) system and the approximate planar motion of vehicles in local areas, inevitably causing additional pose estimation errors. To address this problem, a stereo Visual SLAM system with road constraints based on graph optimization is proposed, called RC-SLAM. Addressing the challenge of representing roads parametrically, a novel method is proposed to approximate local roads as discrete planes and extract parameters of local road planes (LRPs) using homography. Unlike conventional methods, constraints between the vehicle and LRPs are established, effectively mitigating errors arising from assumed six DoF motion in the system. Furthermore, to avoid the impact of depth uncertainty in road features, epipolar constraints are employed to estimate rotation by minimizing the distance between road feature points and epipolar lines, robust rotation estimation is achieved despite depth uncertainties. Notably, a distinctive nonlinear optimization model based on graph optimization is presented, jointly optimizing the poses of vehicle trajectories, LPRs, and map points. The experiments on two datasets demonstrate that the proposed system achieved more accurate estimations of vehicle trajectories by introducing constraints between the vehicle and LRPs. The experiments on a real-world dataset further validate the effectiveness of the proposed system.

**Keywords:** Visual SLAM; road constraint; graph optimization; epipolar constraint; plane feature

## 1. Introduction

With the development of intelligent vehicles, the demand for environmental perception and precise localization is increasing. Visual Simultaneous Localization and Mapping (SLAM), as a vision-based localization and mapping method, holds broad application prospects in intelligent vehicles [1]. Compared to GPS-based methods, stereo Visual SLAM can achieve stable localization and mapping in GNSS-denied scenes, providing precise and stable localization for autonomous driving [1,2]. Compared to Lidar-based SLAM, stereo cameras can similarly acquire precise scales of scenes at lower costs. Simultaneously, stereo cameras offer abundant environmental textures and exhibit more stable performance in structured environments [3].

Generally, stereo Visual SLAM systems assume the camera moves in six degrees of freedom (DoF) space; therefore, the pose estimation is designed within SE(3). However, intelligent vehicles have more stringent motion constraints. Specifically, the motion of the vehicle is constrained by the road, necessitating the vehicle to adhere to the road, resulting in a degradation of its DoF [4]. Consequently, in practical applications, this assumption of 3D space pose estimation conflicts with the approximate planar motion of the vehicle, inevitably causing additional pose estimation errors [5,6].

To address the aforementioned problems, the most direct approach is adding additional constraints to limit the DoF of the system. There are two methods for adding constraints. One method is to integrate additional sensors into the system for data fusion [7]. For instance, fusing a camera with an Inertial Measurement Unit (IMU) utilizes the inertial data from the IMU to further constrain the pose, thereby improving the estimation accuracy of the pose. Typical solutions include OKVIS [8], VINS-Mono [9], among others. However, such methods, when applied to ground vehicles, are affected by factors like uniform speed linear motion or start-stop motion, which degrade IMU observability, subsequently reducing the overall system performance [10–12]. The second method is to use the prior information that the vehicle adheres to the road, introducing constraint relationships between the road and the vehicle to enhance the accuracy of pose estimation without adding sensors [13–15]. However, this method initially requires a parameterized model that accurately represents the road. Given the difficulty in directly measuring the road through sensors [14], the road is often assumed to be a single infinite plane [16], or road parameters are indirectly obtained from low-dimensional features, such as feature points [13] or lines [17].

The current road modeling methods based on the assumption of the infinite plane have been widely applied in indoor scenes, effectively enhancing localization accuracy. However, in outdoor environments, the infinite plane cannot accurately represent the road manifold, and incorrect assumptions might even lead to additional system errors [4]. Therefore, in road scenes, methods based on feature point fitting are commonly used to express the road manifold [18–20]. This method fits the road into a planar model [18,19] or a curved surface model [20], utilizing the fitted road model to constrain the pose of the vehicle. However, road feature points are influenced by low-texture and self-similarity. During the depth recovery of road feature points using stereo disparity, compared to non-road feature points, there is greater depth uncertainty. Consequently, the spatial accuracy of ground feature points is lower, making them unsuitable for direct use in vehicle pose estimation and road model fitting [18,21,22].

This paper proposes an optimization-based stereo Visual SLAM system combined with road constraints, focusing on two key aspects: maximizing the utilization of road features and incorporating vehicle movement on the road. Initially, a method employing homography of local road planes (LRPs) to extract parameters of local roads is proposed. This method approximates the local road as discrete planes, leveraging 2D-2D matching results of road features from previous keyframes to estimate the LRPs of the current keyframe using homography. As this process does not rely on depth information of road features, it circumvents the uncertainty caused by stereo feature matching on road features. By explicitly establishing constraints between vehicle poses and roads, errors arising from the six DoF motion assumptions of vehicles are minimized without any additional sensors. Subsequently, to avoid depth uncertainty when utilizing road feature points, reprojection constraints for non-road feature points and epipolar constraints for road feature points are applied to estimate the motion of the vehicle jointly. Finally, a nonlinear optimization model based on graph optimization is developed. This model joint optimizes vehicle trajectories, LRPs, and map points, thus enhancing the accuracy and robustness of the system.

There are four contributions in this paper:

1. A tightly coupled graph optimization framework is proposed, where explicit constraints between the vehicle and local road planes are established. This framework jointly optimizes the poses of vehicle trajectories, Local Road Planes (LRPs), and map points;
2. To mitigate the impact of depth uncertainties in road features on the estimation of the local road plane, a method using homography is proposed to extract local road plane parameters by leveraging the 2D-2D matched road feature points from previous keyframes to enhance the accuracy of local road plane estimation;

3. A motion estimation method is proposed for road scenes. It employs epipolar constraints for estimating rotation with road feature points to prevent the influence of depth errors and reprojection constraints for estimating both rotation and translation with non-road feature points. The joint optimization through bundle adjustment is used to enhance the robustness and precision of motion estimation;

4. A full SLAM system is proposed that can establish a global map containing map points and local road planes. Extensive validation through multiple datasets and real-world experiments demonstrates the superior performance of the proposed system over state-of-the-art Visual SLAM and Visual-inertial SLAM methods specifically in road scenes.

The rest of the paper is organized as follows: In Section 2, related background research works are discussed. Notations and different plane models are proposed in Section 3. The overview of the entire system and its individual modules are presented in Section 4. Section 5 details the experimental setup and experimental results with result analysis. Finally, the conclusions are given in Section 6.

## 2. Related Work

This paper focuses on the application of Visual SLAM in intelligent vehicles, with a specific emphasis on constraints related to roads and vehicles. Consequently, we have categorized the related work into two parts: the application of Visual SLAM systems in intelligent vehicles and SLAM systems with ground constraints.

### 2.1. Application of Visual SLAM Systems in Intelligent Vehicles

Intelligent vehicles require precise localization and mapping across various scenes. Visual SLAM presents a promising solution. However, it faces challenges such as large-scale scenes, numerous dynamic objects, intense lighting variations, and rapid movements [2]. To address the emerging challenges posed by intelligent vehicles, scholars have conducted research from multiple perspectives including the front-end, back-end, and vision-based multi-sensor fusion [1].

In the front-end, two common methods are the feature-based method [21,23–26] and the direct method (including the semi-direct method) [27–29]. ORB-SLAM2 [24] is a classic feature-point-based Visual SLAM system that estimates camera motion based on feature point extraction, matching, and optimization of the reprojection error. OV$^2$SLAM [26] utilizes LK optical flow to replace ORB descriptors for feature matching, reducing computational load in feature extraction, and thus offering higher real-time performance. SOFT2 [21] and MOFT [22] establish constraints between feature points and epipolar lines, mitigating the impact of depth uncertainty on pose estimation, resulting in improved accuracy and robustness. The direct method estimates camera motion based on pixel grayscale, optimizing photometric errors. Compared to feature-based methods, it does not require the calculation of keypoints and descriptors, presenting advantages in computational speed and the ability to construct dense maps. Representative approaches include SVO [27], LSD-SLAM [28], and DSM [29]. However, the direct method relies on the grayscale constancy assumption and is sensitive to changes in lighting conditions, posing challenges in its application in intelligent vehicles [3].

The back-end receives camera poses and spatial feature points from the front-end and optimizes them to obtain accurate and globally consistent poses and the map. Currently, the back-end can be categorized into filter-based methods [30,31] and optimization-based methods [21,24]. Filter-based methods consist of two stages: state prediction and state correction. These methods first predict the states of vehicles and maps using prediction models and control inputs. Subsequently, they correct the predicted states using sensor measurement. Representative systems include Extended Kalman Filter (EKF) based [30,31] and Multi-State Constraint Kalman Filter (MSCKF) based. In road scenes, due to the high complexity, a large number of features lead to a quadratic growth in state variables, diminishing the real-time advantages of filter-based [1]. Currently, optimization-based

methods, typically represented by graph optimization, consider vehicle poses and features as optimization variables, establishing constraints between vertexes as edges in a graph and optimizing the graph to obtain accurate vehicle poses and the map. However, this approach offers higher precision at the cost of increased computational expenses [21]. Given the real-time requirements of intelligent vehicles, methods like sparse matrix decomposition, sliding windows, and local maps are applied in graph optimization. Higher precision and acceptable real-time performance have made optimization-based methods the current mainstream [32].

To further enhance system robustness, multi-sensor fusion methods based on visual sensors have garnered significant attention [2]. Due to the complementarity of IMU and camera, VI-SLAM systems [9,12,15,33,34] fuse IMU preintegration and Visual Odometry(VO) through graph optimization or filter to obtain more accurate and robust camera poses and the map. However, VI-SLAM faces challenges in system initialization and observability due to vehicle dynamics constraints and road constraints [15,35]. Lidar provides precise structural information, while cameras capture abundant environmental texture. In recent years, there has been a wave of odometry and SLAM systems that fuse camera, Lidar, and IMU [34,36,37]. These systems aim to attain more accurate state estimation in complex and dynamic environments. However, these systems also face challenges due to the computational complexity caused by fusing multiple data.

*2.2. SLAM Systems with Ground Constraints*

Due to the inevitable constraints imposed by the ground on vehicles, many researchers have proposed SLAM methods that integrate road constraints. Wei et al. [38] proposed a Lidar SLAM system designed for indoor parking lots. This system utilizes ground constraints, representing the ground as plane features to enhance constraints in the vertical direction, thus reducing vertical pose drift. Wu et al. [35] demonstrated the impact of degenerate motion on the Visual-inertial Odometry (VIO). Addressing this problem, they proposed to integrate random plane constraints into the VIO improving pose accuracy with wheel odometer measurements. In [39], a pose parameterization method named SE(2)-constrained SE(3) poses, which allows 3-D transformations constrained by 2D planar motion with small perturbations, was proposed. The authors suggested that this method maximally accommodates real-world scenes in indoor navigation settings. Zheng et al. [40] proposed a VO based on a wheel odometer and camera, directly parameterizing the pose using SE(2) and considering disturbances beyond SE(2) as visual measurement errors. In indoor scenes, this system demonstrates superior accuracy and robustness. However, these methods primarily focus on indoor, parking lots, or factories with ground planes, which limits their applicability in complex road scenes.

In road scenes, integrating road constraints into SLAM systems similarly allows for better estimation of poses in three-dimensional space for vehicles [6]. Wen et al. [41] proposed to use the absolute position of road planes fitted from Lidar points to constrain the vertical pose estimation of vehicles. Additionally, the plane normal is used to constrain pose drift. In [19,20], the constraint between the vehicle and the road is utilized to establish the vehicle-ground geometry and recover the scale for monocular Visual Odometry. In [19], parameters of the discrete plane are estimated using feature points in the Region of Interest (ROI), while the road is modeled as a quadratic polynomial in [20]. A quadratic polynomial is similarly used to parameterize the road manifold in both [14,15]. In [14], pose integration is performed using measurements from both the IMU and the wheel odometer, which are fused into the proposed representation of the road manifold. However, in [15], the six DoF pose integration based on the road manifold is reliant on measurements from the wheel odometer. In [42], B-splines are utilized to represent a continuous and smooth trajectory of the vehicle. This representation can also be treated as a method to parameterize the road model. The utilization of this trajectory effectively enhances the accuracy and robustness of monocular VO. When employing high-dimensional models like polynomials [14,15,20] and B-splines [42,43] to represent the road, initializing parameters becomes challenging.

Moreover, when the shape of the road changes rapidly, the parameter update process struggles to converge quickly. A method similar to this paper is presented in [4], which indirectly acquires a road model by utilizing spatial road feature points from the camera. It fits these points into a sequence of local planes with varying slopes and maximizes the usage of road constraints based on random constraints between the camera and discrete local planes.

Compared to the mentioned methods, the proposed system focuses on utilizing the road to constrain the SLAM system from two perspectives: "maximizing the use of road features" and "vehicle move on the road". Initially, the proposed system utilizes the matched 2D road feature between consecutive keyframes to establish epipolar constraints, achieving a more accurate estimation of rotation. Subsequently, the proposed system employs homography to estimate the LRPs of the current frame by using observations from previous keyframes to impose road constraints on the vehicle. Both contribute to the accuracy and robustness of vehicle pose estimation.

## 3. Preliminaries

### 3.1. Notation

The notations that are used throughout the paper are defined first. $(\cdot)^W$ represents the world frame, $(\cdot)^C$ represents the camera frame, $(\cdot)^B$ represents the body frame of vehicle, and the body frame is located at the projection on the road of the center of the rear axle. The Euclidean transformation between the world frame and $k$th camera frame can be represented as follows:

$$T_W^{C_k} = \begin{bmatrix} R_W^{C_k} & t_W^{C_k} \\ \mathbf{0}^{\mathrm{T}} & 1 \end{bmatrix} \in \mathrm{SE}(3) \mid R_W^{C_k} \in \mathrm{SO}(3), t_W^{C_k} \in \mathbb{R}^3, \tag{1}$$

where $T_W^{C_k}$ is transformation matrix from the world frame to $k$th camera frame, $R_W^{C_k}$ is rotation matrix, $t_W^{C_k}$ is translation vector. Using the transformation matrix $T_W^{C_k}$, 3D landmarks $P_i^W$ can be converted from the world frame to $k$th camera frame: $P_i^{C_k} = \left( T_W^{C_k} P_i^{W'} \right)_{[1:3]} = R_W^{C_k} P_i^W + t_W^{C_k}$, where $P'$ is the homogeneous form of $P$, and $P_i^{C_k}$ is 3D landmarks in $k$th camera frame. The Euclidean transformation includes rotation and translation, first 3D landmarks $P_i^W$ in the world frame need to be rotated by rotation matrix $R_W^{C_k}$, and then translation vector $t_W^{C_k}$ need to be added to obtain the translated 3D marks $P_i^{C_k}$. $K$ is the intrinsic parameters of the camera, these parameters need to be obtained in advance through calibration.

### 3.2. Road Plane Models

Local roads can be approximated as discrete planes. In this paper, the plane is parameterized by Hesse Form (HF). The expression of a plane is $\pi = \begin{bmatrix} n^{\mathrm{T}} & d \end{bmatrix}^{\mathrm{T}}$. Point $P$ lying in the plane should satisfy $n^{\mathrm{T}} P = d$, where $n \in \mathbb{R}^3$ is the unit vector, $d$ is the distance from the plane to the origin of the frame. Using HF allows for convenient transformation of the plane between different frames. As shown in Figure 1, the transformation relationship of the plane between the world coordinate system and the camera coordinate system can be represented as follows:

$$\begin{bmatrix} n^W \\ d^W \end{bmatrix} = \begin{bmatrix} R_{C_k}^W & 0 \\ \left( -t_{C_k}^W \right)^{\mathrm{T}} & 1 \end{bmatrix} \begin{bmatrix} n^{C_k} \\ d^{C_k} \end{bmatrix}, \tag{2}$$

where $\pi^W = \begin{bmatrix} \left( n^W \right)^{\mathrm{T}} & d^W \end{bmatrix}^{\mathrm{T}}$ is the parameters of plane in world frame $O^W$, $\pi^{C_k} = \begin{bmatrix} \left( n^{C_k} \right)^{\mathrm{T}} & d^{C_k} \end{bmatrix}^{\mathrm{T}}$ is the parameters of plane in camera frame $O^{C_k}$.

HF uses four parameters to parameterize a plane, yet a plane in three-dimensional space only has three DoF, leading to the plane being over-parameterized. When using the Gauss-Newton optimization, over-parameterization of the plane leads to the computed Hessian matrix during optimization not being full rank, thereby rendering it non-invertible. To solve this problem, inspired by the Closest Point (CP) in [44], we proposed a method during the optimization process that utilizes the Inverse Closest Point (ICP) to parameterize plane, which parameterizes the plane as $\Pi = \boldsymbol{n}/d$. The transformation relationship between ICP and HF can be expressed as:

$$\left[\begin{array}{c} \boldsymbol{n} \\ d \end{array}\right] = \left[\begin{array}{c} \Pi/\|\Pi\| \\ 1/\|\Pi\| \end{array}\right].$$ (3)

The main advantage of Inverse Closest Point is that it parameterizes the plane with only three parameters, avoiding the problem of over-parameterization. The error model during the parameter update in the optimization process is also a simple additive model. Combining the advantages of HF and ICP, the plane parameters are stored in the form of HF for ease of frame transformations. When optimization of the plane is required, the representation is switched to ICP.

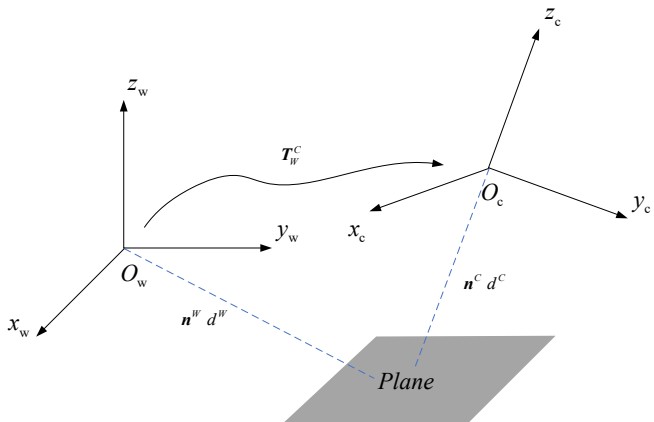

**Figure 1.** The transformation relationship of the plane from world frame to camera frame.

## 4. Proposed Method

### 4.1. System Overview

The pipeline of the proposed system is shown in Figure 2. The system consists of three main parts: the front-end, local road modeling, and back-end. There are two methods to obtain the road area, one is using the method of clustering [45,46], and another is using semantic segmentation [47]. The system takes stereo images and left semantic images with masks of road obtained through a semantic segmentation network [47] as input. The output includes the vehicle poses and a global map containing map points and local road planes.

Similar to many Visual SLAM [24,25], the front-end processes stereo images in a sequence of feature extraction, stereo matching, inter-frame feature matching, and motion estimation. Keyframe selection relies on pose estimation and inter-frame co-visibility. Within the front-end, features are categorized into road features and non-ground features based on the semantic image. A more stringent inter-frame feature matching approach is proposed specifically for the road features.

The local road model operates in parallel with the back-end. The local road model is modeled as discrete planes in this part. Whenever a new keyframe is generated, the local road model models the corresponding local road plane for the keyframe. During the plane fitting process, measurements of the local road and pose estimations from previous keyframes are utilized to compute the relevant homography for the plane parameters of the new keyframe.

The back-end consists of two parts: Local Bundle Adjustment (LBA) and Loop Correction. Within LBA, road constraints on the vehicle are enhanced from two perspectives. The vehicle trajectory, local road planes, and map points are jointly optimized in the LBA, leading to more accurate pose estimations and maps. Loop Correction executes when the system detects loop closures. It performs global optimization on the vehicle trajectory, local road planes, and map points using loop closure constraint, rectifying accumulated pose drift within the system.

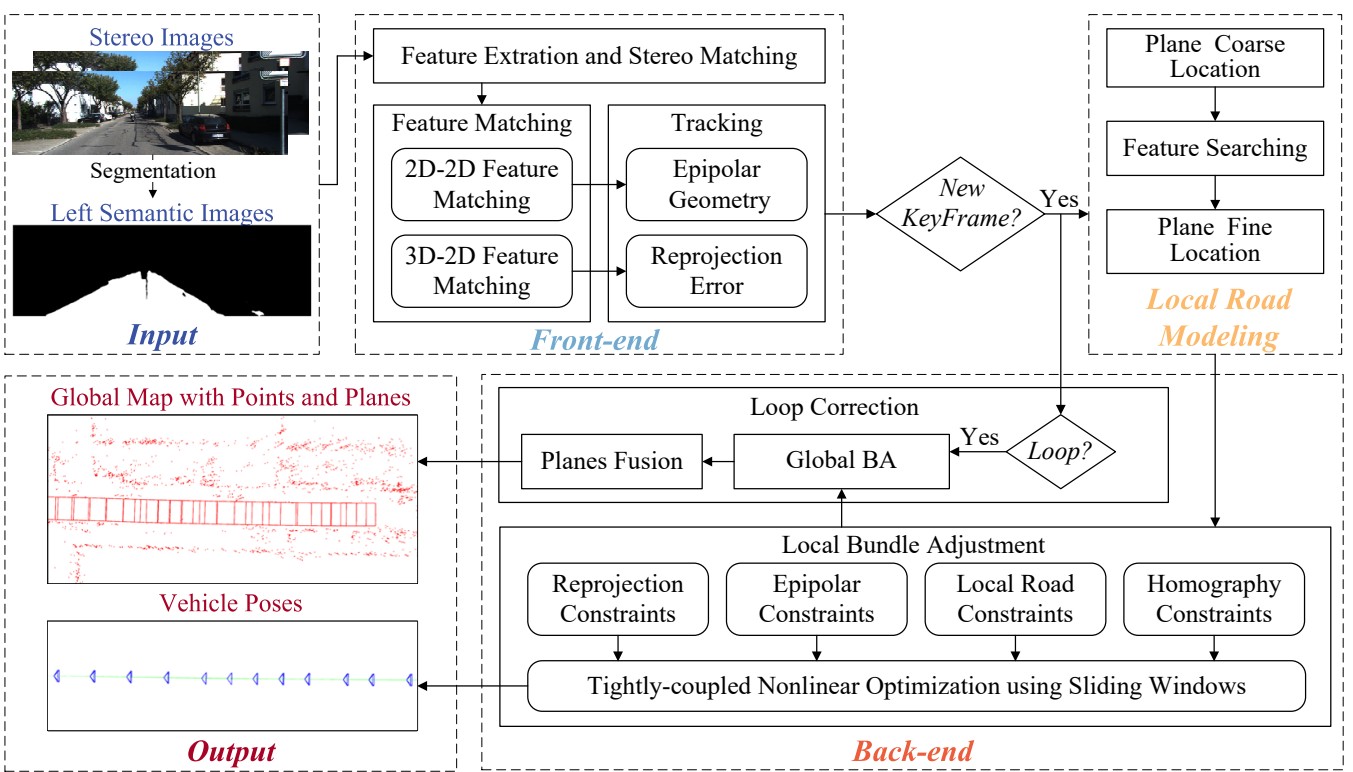

**Figure 2.** The pipeline of the proposed system.

### 4.2. Front-End

In the front-end, feature points are extracted from the stereo images. Subsequently, feature points from the left and right images are stereo-matched, and the inter-frame matched feature points are used to calculate the camera's pose changes.

#### 4.2.1. Feature Extraction and Stereo-Matching

For the input, an image pyramid is initially constructed for both left and right images to ensure feature scale invariance. To ensure an even distribution of feature points across the entire image pyramid, each level of the image pyramid is subdivided into multiple $60 \times 60$ grids. Within each grid, the ORB features and descriptors [24] are extracted until the number of feature points in each grid reaches the preset threshold or no qualifying features are found within the grid. After completing the extraction of image features, the depth of each feature point is recovered based on the stereo-matching results of feature points between the left and right images. The stereo-matching process between the left and right images involves epipolar line searches within the same pixel row. Subsequently, sub-pixel optimization is applied to attain more accurate depth for feature points.

As mentioned earlier, compared to other features, stereo-matching in road features often leads to larger disparity errors, increasing the uncertainty in depth estimation for road features. Moreover, as shown in Figure 3, road feature points may be extracted from shadows on the road, which lack temporal invariance and are unsuitable for inclusion in the map. Due to these two reasons, road feature points are not included in the map in the proposed system. Instead, the local road planes are estimated using 2D road features, integrating stable plane features into the map.

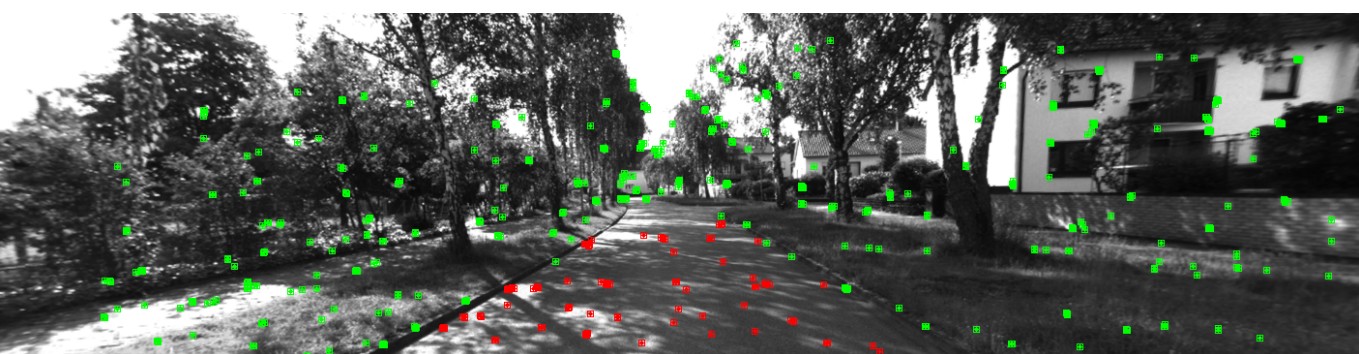

**Figure 3.** Detected ORB features in frame 1839 of KITTI-360 sequence 00. Road features are marked in red, and others are marked in green. Most road features are located at the edge of the shadow.

### 4.2.2. Feature Tracking

For non-road feature points, feature matching between frames uses reprojection for 3D-2D feature tracking. The 3D feature points from the previous frame are projected onto the image of the current frame. An association gate centered around the reprojection point with a fixed radius is set up, and features falling within this gate are matched to establish the inter-frame feature points correspondence.

However, for features on the road, their large depth error causes deviations in the reprojected feature point, leading to decreased matching success rates and accuracy. To obtain accurately matched road feature points, a 2D-2D matching approach is proposed for the road features in consecutive left images. The pseudo-code for the road feature matching process is shown in Algorithm 1. In lines 1–9, the coarse matching of road feature points is executed based on the distance of descriptors. By computing the Hamming distance between descriptors of road features from consecutive frames, the feature pairs with descriptors having distances smaller than the reset threshold are stored as matching candidates along with their corresponding distances. In lines 10–24, a more refined matching process is performed for road feature points. Due to the self-similarity of the road, in the coarse matching process, a feature point often matches with multiple feature points in the next frame. To achieve a globally optimal match, the Hungarian matching method was adopted, using the reciprocal of the feature point distance as the weight, obtaining the globally minimum-cost fine matching results. To further eliminate outliers, in lines 25–40, the Random Sample Consensus (RANSAC) randomly selects the minimum sample set to estimate the initial epipolar geometry model. Through epipolar constraints, all matching pairs are judged to conform to the epipolar geometry relationship, classifying them as inliers or outliers. If the ratio of inliers meets the preset requirements, all outliers are removed. All inliers are used for subsequent processes.

---

**Algorithm 1:** 2D Road Features Matcher

---

    **Input:** ORB Features in road from two consecutive camera frames $F_k$, $F_{k+1}$.
    **Output:** Matched road features $\left(F_k^*, F_{k+1}^*\right)$.
    `// Coarse matching of features based on descriptor distance`

**1**  **for** $i = 1 : |F_k|$ **do**
**2**     **for** $j = 1 : |F_{k+1}|$ **do**
            `// Calculate the Hamming distance of the descriptor`
**3**         **if** hanming_distance $\left(F_{k,i}, F_{k+1,j}\right) >$ preset_distance **then**
**4**             $features\_distance \leftarrow$ hanming_distance $\left(F_{k,i}, F_{k+1,j}\right)$;
**5**         **end**
**6**     **end**
**7**  **end**
    `// Finer matching of features based on weighted Hungarian algorithm`
**8**  $C = $ cost_matrix($features\_distance$);
**9**  **if** numRows(C) != numRow(C) **then**
**10**     $C = $ square_matrix($C$);
**11**  **end**
**12**  **for** $row = 1 : |C|$ **do**
**13**     **for** $col = 1 : |C|$ **do**
**14**         **if** $potentialRow[row] + potentialCol[col] - C[row][col] < minCost$ **then**
**15**             $matches[col] = row$;
**16**             $minCost = potentialRow[row] + potentialCol[col] - C[row][col]$;
**17**         **end**
**18**         $potentialRow[row] += minCost$;
**19**     **end**
**20**     $potentialCol[Col] -= minCost$;
**21**  **end**
    `// Removing outliers with RANSAC`
**22**  **for** $m = 1 : maxIterations$ **do**
**23**     **if** isEpipolarConstraintSatisfied($matches, possibleModel, distanceThreshold$) **then**
**24**         increment($inliers$) ;
**25**         $inlierIndices.append(n)$;
**26**     **end**
**27**     **if** $inliers > bestInliers$ **then**
**28**         $bestInliers = inliers$ ;
**29**         $\left(F_k^*, F_{k+1}^*\right) \leftarrow inlierIndices$ ;
**30**     **end**
**31**  **end**
**32**  **return** $\left(F_k^*, F_{k+1}^*\right)$

---

### 4.2.3. Motion Estimation

Due to the accurate spatial information of non-ground feature points, and conversely, the poor accuracy of ground feature points in spatial information, when estimating motion between frames, 3D-2D reprojection constraints and 2D-2D epipolar constraints are separately established for these two types of feature points. For non-ground feature points, re-projection error can be used to construct constraints,

$$e_{\text{reproj}} = u_l^{C_{k+1}} - \frac{1}{S_{P_l}} K \left( R_{C_k}^{C_{k+1}} P_l^{C_k} + t_{C_k}^{C_{k+1}} \right), \tag{4}$$

where $P_l^{C_k}$ and $u_l^{C_{k+1}}$ are the matched 3D feature and 2D feature in two frames, $S_{P_l}$ is the depth of $P_l^{C_k}$. This constraint can simultaneously estimate the rotation and translation of the vehicle. For road feature points, only 2D-2D matched features in consecutive frames are used to construct epipolar constraints, aiming to avoid errors caused by the depth uncertainty of feature points. The epipolar constraint describes the constraints formed by the 2D feature and the camera optical center when the same feature is projected onto images from two different perspectives under the projection model. When constructing epipolar constraints, the epipolar lines can be represented as:

$$l = \left(t_{C_k}^{C_{k+1}}\right)^{\wedge} R_{C_k}^{C_{k+1}} p_j^{C_k},　　　　　　　　　(5)$$

where $p_j^{C_k}$ is the observation of road feature point on the normal plane of the $k$th frame, $p_j^{C_k} = (x_j, y_j, 1)^{\mathrm{T}} = K^{-1}(u_j, v_j, 1)^{\mathrm{T}}$.

According to [21], the distance between matched feature points in the $k + 1$th image and the epipolar line is considered as the epipolar error. By adjusting the rotation and translation changes between the two frames, the goal is to minimize this error. The distance from the matched feature point in the $k + 1$th image to the epipolar line constructed based on Equation (5) can be obtained as:

$$e_{\text{epipolar}} = \frac{\left| l^{\mathrm{T}} p_j^{C_{k+1}} \right|}{\|l\|},　　　　　　　　　(6)$$

where $p_j^{C_{k+1}}$ is the matched point of $p_j^{C_k}$ in the $k + 1$th image. Because 2D-2D matching does not involve scale, this constraint is used to estimate only the rotational of the vehicle. In contrast to [21], where the epipolar errors are computed for all points, this constraint to road points is applied to road features, while non-road points continue using reprojection error, which can reduce information loss. The Jacobian of error term with respect to $R_{C_k}^{C_{k+1}}$ is

$$J\left(R_{C_k}^{C_{k+1}}\right) = \left(\frac{p_j^{C_{k+1}\,\mathrm{T}} l \cdot l^{\mathrm{T}}}{\|l\|^3} - \frac{p_j^{C_{k+1}\,\mathrm{T}}}{\|l\|}\right)\left(t_{C_k}^{C_{k+1}}\right)^{\wedge}\left(R_{C_k}^{C_{k+1}} p_j^{C_k}\right)^{\wedge}.　　　(7)$$

Integrating both constraints based on their error functions, a comprehensive optimization problem is formulated in Equation (8), where a nonlinear optimization method is employed to minimize the overall error function. By meticulously handling both ground and non-ground feature points, this differentiated strategy fully utilizes the characteristics of road feature points while enhancing the robustness and precision of motion estimation.

$$E_{ME} = \sum\left\|e_{\text{reproj}}\right\|_{\Sigma_{\text{reproj}}^{-1}} + \sum\left\|e_{\text{epipolar}}\right\|_{\Sigma_{\text{eppipolala}}^{-1}}.　　　　　（8）$$

### 4.3. Local Road Modeling

When a new keyframe is detected in the front-end, it is necessary to estimate the local road where the newly generated keyframe is located. Due to the continuous shape and small gradient changes of roads, they can be approximated as planes within small-scale areas. Therefore, roads can be divided into a series of discrete planes. Based on this analysis, a plane model will be used to represent the local roads in this context.

When the current camera frame is detected as a keyframe, the initial position of the current frame is determined based on the motion estimation results from the front-end. Given the known mounting position of the camera, the projection of the camera onto the road is determined using the extrinsics between the camera and the road. So the road area where the keyframe is located is identified, as shown in the gray region in Figure 4. The size of this region is empirically set to be 6 m in length and 4 m in width.

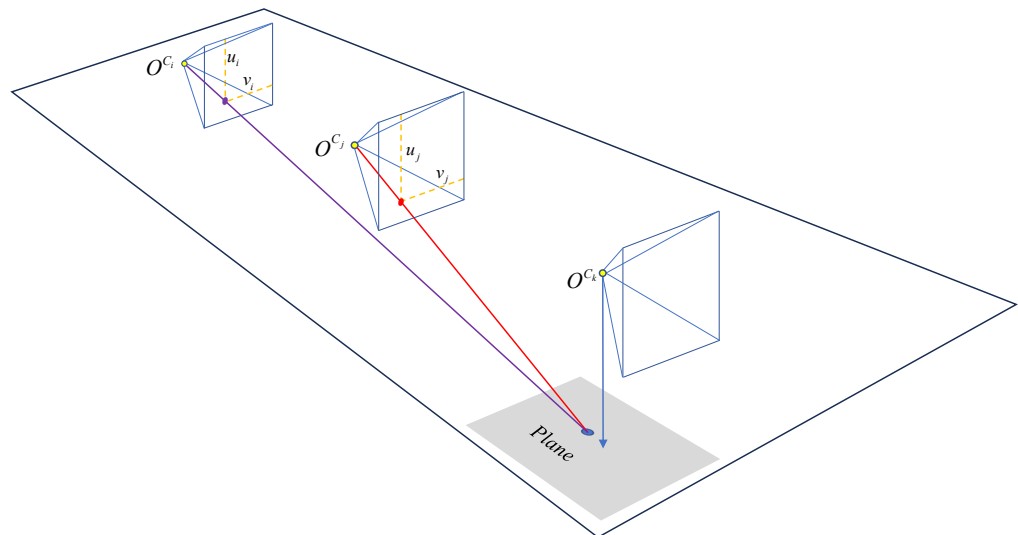

**Figure 4.** The observed relationship of the local road plane. The gray area represents the local road plane to be fitted, while the blue point denotes spatial feature points located on the local road plane. The red point and purple point are, respectively, the projection point of blue points on the imaging planes of previous keyframes $C_i$ and $C_j$.

Once the local road area for the vehicle is determined, the previous keyframes that can observe this plane need to be found. To enhance search efficiency, previous keyframe groups capable of observing this area are determined based on the positions of previous keyframes and the field of view (FOV) of the camera. Subsequently, road feature point pairs located on the local road plane are selected from the previous frame groups, and the two keyframes possessing the most point pairs are identified. It is important to note while selecting road feature points, the 3D information of these points is utilized solely to confirm whether the feature points lie on the plane to be fitted. However, in the subsequent estimation of the road plane, only the 2D information of these road feature points is utilized.

After obtaining the matched feature point pairs as above, the local road plane needs to be fitted next. In contrast to the method outlined in [4], which utilizes 3D feature points for plane fitting, here, only the 2D feature points from previous frames and their matching relationships are used to prevent the influence of depth errors of road points. There is a similar method [48] that uses 2D feature points to construct homography between two keyframes. However, they use this constraint to optimize poses, whereas homography is employed to optimize planes in the proposed method. As shown in Figure 4, for previous keyframes $C_i$ and $C_j$, some road features lie on the local road plane where the current frame is positioned. In other words, there are some matching pairs of road points between these two previous keyframes that fall in the same road plane. According to epipolar geometry, if the points lie on the same plane, they can be constrained using homography:

$$\lambda \boldsymbol{p}_k^{C_j} = \boldsymbol{H} \boldsymbol{p}_k^{C_i}, \tag{9}$$

$$\boldsymbol{H} = \boldsymbol{R}_{C_j}^{C_i} - \boldsymbol{t}_{C_j}^{C_i} \Pi_{C_i}^{\mathrm{T}}, \tag{10}$$

where $\lambda$ is the scale, $\boldsymbol{p}_k^{C_i}$ and $\boldsymbol{p}_k^{C_j}$ is the observation of road feature point on the normal plane of $i$th frame and $j$th frame, $\boldsymbol{H}$ is the homography, $\boldsymbol{R}_{C_j}^{C_i}$ and $\boldsymbol{p}_k^{C_j}$ are the rotation and translation from $i$th keyframe to $j$th keyframe, both of them are known. $\Pi_{C_i}$ is the local road plane in $i$th camera frame. In the established homography constraint, the variables involve the parameters of the plane and the rotation and translation between two frames. Furthermore, given the known pose changes between historical keyframes, the plane parameters can be

extracted based on homography. According to Equation (8), the homography constraint between *k*th matched point pair is shown as:

$$
\lambda \begin{bmatrix} x_j \\ y_j \\ 1 \end{bmatrix} = \begin{bmatrix} h_1 & h_2 & h_3 \\ h_4 & h_5 & h_6 \\ h_7 & h_8 & h_9 \end{bmatrix} \begin{bmatrix} x_i \\ y_i \\ 1 \end{bmatrix}.
\tag{11}
$$

Derived from expanding Equation (11), the homography constraint can be expressed as

$$
\mathbf{0} = \begin{bmatrix} x_i & y_i & 1 & 0 & 0 & 0 & -x_i x_j & -y_i x_j & -x_j \\ 0 & 0 & 0 & x_j & y_j & 1 & -x_i y_j & -y_i y_j & -y_j \end{bmatrix}
$$
$$
\begin{bmatrix} h_1 & h_2 & h_3 & h_4 & h_5 & h_6 & h_7 & h_8 & h_9 \end{bmatrix}^{\mathrm{T}}.
\tag{12}
$$

Accurate matching relationships of road points are obtained from the front-end. Considering the sensitivity of the epipolar constraint to noise, in order to reduce the impact of outliers on plane estimation, the 4-point method is used with RANSAC to select all inliers to compute $\boldsymbol{H}$. The parameters of plane $\Pi_{C_i}$ are also calculated as the initial value for subsequent optimization after obtaining the $\boldsymbol{H}$. Similar to [18], the homography error can be expressed as

$$
\boldsymbol{e}_{\text{homography}} = \boldsymbol{p}_k^{C_j} - \left( \boldsymbol{R}_{C_i}^{C_j} - \boldsymbol{t}_{C_i}^{C_j} \Pi_{C_i}^{\mathrm{T}} \right) \boldsymbol{p}_k^{C_i}.
\tag{13}
$$

Given that the rotation $\boldsymbol{R}_{C_j}^{C_i}$ and translation $\boldsymbol{t}_{C_j}^{C_i}$ between *i*th keyframe and *j*th keyframe have been estimated in previous sliding window, optimization is solely performed on the plane. The Jacobian of error term with respect to $\Pi_{C_i}$ is

$$
\boldsymbol{J}\left( \Pi_{C_i} \right) = \left( \left( \boldsymbol{p}_k^{C_i} \right)^{\mathrm{T}} \otimes \boldsymbol{I}_{3\times3} \right) \left( \boldsymbol{I}_{3\times3} \otimes \boldsymbol{t}_{C_i}^{C_j} \right),
\tag{14}
$$

where $\otimes$ is the Kronecker product, $\boldsymbol{I}_{3\times3}$ is the identity matrix.

### 4.4. Back-End

When keyframes are detected in the front-end, the keyframes will pass to the back-end. In the back-end, a Local Bundle Adjustment (LBA) is performed to optimize all keyframes within the sliding window along with the corresponding points and local road planes. Simultaneously, a check is conducted to determine whether the current keyframe exhibits a loop closure with previous keyframes stored in the map. If loop closure conditions are satisfied, a global BA is executed for loop closure correction.

#### 4.4.1. Local Bundle Adjustment

In LBA, the optimized variables include the poses of all keyframes in the sliding window, as well as all the map points and local road planes corresponding to these keyframes. Figure 5 shows the factor graph of the proposed Local Bundle Adjustment with points and road planes. The LBA incorporates four distinctive constraint types: reprojection constraints, linking keyframes to non-road points; epipolar constraints, connecting keyframes to road points; constraints associating the vehicle with Local Road Planes (LRPs); and homography constraints between preceding keyframes and LRPs. These diverse constraints form the basis for constructing error functions, facilitating the concurrent optimization of poses, landmarks, and LRPs. The error functions are represented in the least squares form and iteratively solved using the Gauss-Newton method from the G$^2$osolver [49], with a maximum iteration limit set to 10. The optimization goal in LBA is to minimize the following loss function:

$$
E_{LBA} = \sum_{i,j} \left\| \boldsymbol{e}_{\text{reproj}}^{i,j} \right\|_{\Sigma_{i,j}^{-1}} + \sum_{i,l} \left\| \boldsymbol{e}_{\text{epipolar}}^{i,l} \right\|_{\Sigma_{i,l}^{-1}}
$$
$$
+ \sum_{k,m} \left\| \boldsymbol{e}_{\text{homography}}^{k,m} \right\|_{\Sigma_{k,m}^{-1}} + \sum_{i,k} \left\| \boldsymbol{e}_{\text{KF-LRPs}}^{i,k} \right\|_{\Sigma_{i,k}^{-1}},
\tag{15}
$$

where $e_{\text{reproj}}^{i,j}$ is the projection error, $e_{\text{epipolar}}^{i,l}$ is the epipolar error, $e_{\text{homography}}^{k,m}$ is the homography error, $e_{\text{KF-LRPs}}^{i,k}$ is the error between keyframe and LRPs. $\Sigma_{i,j}^{-1}$, $\Sigma_{i,l}^{-1}$, $\Sigma_{k,m}^{-1}$ and $\Sigma_{i,k}^{-1}$ are the information matrices corresponding to the four types of errors. The $e_{\text{reproj}}$ and $e_{\text{epipolar}}$ are similar to errors in the Section 4.2.3, and the $e_{\text{homography}}$ is described in Section 4.3.

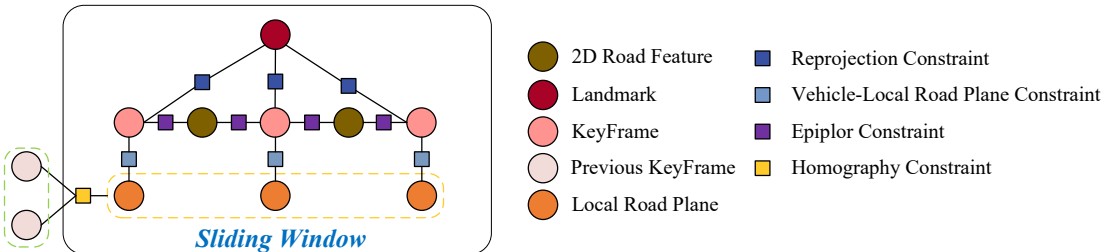

**Figure 5.** The local road planes before and after loop closure correction.

The $e_{\text{KF-LRPs}}$ is used to constrain the position of the vehicle through the local road plane. Based on the attachment between the vehicle and local road planes, where the vehicle should be in complete contact with the road, constraints can be established between the camera and the road plane. In contrast to [4], the proposed method diverges in its approach by not relying on the direct interaction between the road plane and the four wheels to establish constraints. Instead, it leverages the extrinsics between the camera and the vehicle's body frame. This technique involves transforming the camera pose to align with the body frame, establishing a singular-point constraint with the road plane. In contrast to the four-point constraints, the single-point constraint aims to minimize the impact on the system caused by errors in the plane normal and changes in extrinsics. The error is represented as follows:

$$e_{KF-LR}^{i,k} = e\left(\boldsymbol{T}_W^{C_i}, \Pi_k^{\text{T}}\right) = \frac{\Pi_k^{\text{T}}\left(\boldsymbol{T}_W^{C_i}\right)^{-1}\left(\boldsymbol{t}_B^{C_i}\right)'}{\|\Pi_k\|} - \frac{1}{\|\Pi_k\|}, \tag{16}$$

where $\boldsymbol{T}_W^{C_i}$ is the pose of $i$th keyframe, $\left(\boldsymbol{t}_B^{C_i}\right)'$ is the homogeneous form of the origin of the vehicle body frame in the camera frame, $\Pi_k^T$ is the $k$th plane in the map, which corresponds to the local road plane of the vehicle in the $i$th keyframe. The Jacobian of error term with respect to $\boldsymbol{T}_W^{C_i}$ is

$$J\left(\boldsymbol{T}_W^{C_i}\right) = \frac{\Pi_k^{\text{T}}\left[\boldsymbol{R}_W^{C_i\ \text{T}}\left(\boldsymbol{t}_B^{C_i}\right)^{\wedge} - \boldsymbol{R}_W^{C_i\ \text{T}}\right]}{\|\Pi_k\|}, \tag{17}$$

where $\left(\boldsymbol{t}_B^{C_i}\right)^{\wedge}$ is the skew symmetric matrix of $\boldsymbol{t}_B^{C_i}$. The Jacobian of error term for $\Pi_k$ is

$$J\left(\Pi_k^{\text{T}}\right) = -\frac{\Pi_k^{\text{T}}\left(\left(\boldsymbol{T}_W^{C_i}\right)^{-1}\left(\boldsymbol{t}_B^{C_i}\right)'\right)\Pi_k^{\text{T}}}{\|\Pi_k\|^3} + \frac{\left(\left(\boldsymbol{T}_w^{c_i}\right)^{-1}\left(\boldsymbol{t}_B^{C_i}\right)'\right)^{\text{T}}}{\|\Pi_k\|} + \frac{\Pi_k^{\text{T}}}{\|\Pi_k\|^3}. \tag{18}$$

### 4.4.2. Loop Correction

Loop Correction and LBA run in parallel in the back-end. When a new keyframe is detected, the proposed system, similar to Visual SLAM [24,25], employs loop closure detection using a Bag-of-Words (BoW) model based on DBoW2 [49] to identify if the current frame forms a loop with previous keyframes in the map. If an accepted loop closure is detected, a global Bundle Adjustment (BA) is executed to rectify accumulated drift within the loop. During the global optimization, points, keyframes, and local road planes are simultaneously optimized. Figure 6 shows local road planes before and after loop correction.

The poses of keyframes and local road planes are simultaneously adjusted to achieve a more accurate map.

After loop closure occurs, the local road planes might be overlapping. The numerous repeated planes in the map lead to unnecessary consumption of computational resources and storage space. The keypoint of plane fusion is to determine whether two planes should be merged. Excessive fusion may result in planes that do not conform to the shape of the road, while insufficient fusion fails to resolve plane overlaps. The commonly used Intersection over Union (IoU) from object detection tasks is used to calculate the overlap ratio of planes, denoted as $IoU = A \cap B / A \cup B$. By considering the distance between the center positions of planes, it is possible to quickly assess the likelihood of plane overlap. For two overlapping planes, if their IoU is greater than 0.5, two planes will be fused. During the fusion process, the matching relationship of feature points obtained from local road modeling is utilized to recompute the parameters of the fused plane, and the size of the fused plane is the Union of the two planes.

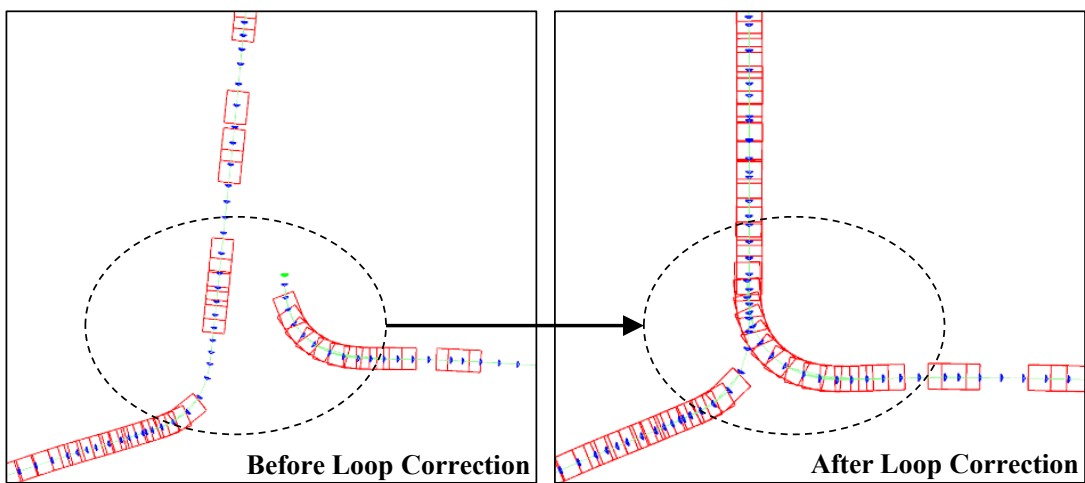

**Figure 6.** The factor graph of the proposed Local Bundle Adjustment with points and road planes.

## 5. Experiments

RC-SLAM was evaluated using two publicly available datasets, KITTI-360 [50] and KITTI [51], along with a real-world dataset collected by a physical test platform. The selection of these datasets was carefully considered, taking into account factors such as the diversity and complexity of scenes, as well as the widespread usage of the datasets. For reference, the performance of open-source Visual SLAM systems: ORB-SLAM2 [24], OV$^2$SLAM [26], and the Visual-inertial SLAM system ORB-SLAM3 [25] were also tested on the aforementioned datasets. The ablation experiments were performed on the KITTI-360 dataset to evaluate the two proposed constraints within RC-SLAM. To minimize the impact of randomness in each system, each system was consecutively run five times on every sequence of the dataset. It is important to note that both the proposed system and three open-source systems were implemented on a computer equipped with an Intel i7-11700 CPU at 3.6 GHz.

The systems were evaluated using two metrics: Absolute Trajectory Error (ATE) $t_{ate}$ [52] and Relative Pose Error (RPE) [53]. ATE assesses the global consistency of the system by comparing the root mean square error (RMSE) between the estimated trajectory and the ground truth. RPE consists of Relative Translation Error $t_{rel}$ and Relative Rotation Error $r_{rel}$. It describes the local accuracy within fixed time intervals, suitable for assessing the drift of the system. It is important to note that alignment between the coordinate systems of each system and the ground truth is necessary before evaluation. Here, the Umeyama algorithm [54] was used to process the aforementioned data.

### 5.1. KITTI-360 Dataset

The KITTI-360 dataset offers nine sequences with Ground Truth data. These sequences encompass various scenes, including low-speed driving in urban and high-speed driving on busy highways. The dataset contains multiple sensor data, including a stereo color camera operating at 10 Hz with a baseline of 0.6 m, two fish-eye cameras with a 180-degree FOV, a 64-line Lidar, and an OXTS3003 GPS/IMU Unit. Rectified stereo images and the provided Ground Truth from the dataset were used in experiments. It is noteworthy that the Ground Truth in this dataset is obtained through large-scale optimization using OXTS measurements, laser scans, and multi-view images, resulting in more accurate poses. Compared to the KITTI dataset, the Ground Truth in KITTI-360 is considered more accurate and reliable; thus, the ablation experiments were performed on this dataset. However, ground-truth poses are not available for each frame in all sequences. Some image frames lack corresponding Ground Truth, such as the first frame of each sequence. Hence, seven fragments with continuous Ground Truth from 7 sequences were selected to evaluate each algorithm. The corresponding camera frames at the beginning and end of each fragment were specified. For a direct comparison, all systems have closed-loop correction.

Table 1 shows the ablation experiments conducted by gradually adding the proposed methods for comparison. The proposed system was divided into three parts. VSLAM serves as the baseline, a basic Visual SLAM based solely on stereo images without any treatment of road features. VSLAM+EG integrates road features onto VSLAM and enforces 2D-2D epipolar constraints on road feature points, while non-ground feature points still adhere to 3D-2D reprojection constraints. VSLAM+EG+LRC represents the complete proposed system, which further performs local road planes to add constraints between road planes and vehicles based on VLSAM+EG. The results indicate the improvement in the $r_{rel}$ after adding epipolar geometry constraints. This improvement is due to mitigating the influence of depth uncertainty of road feature points on rotation estimation by employing epipolar constraints. However, 2D road feature points cannot obtain the scale, so epipolar geometry constraints cannot directly constrain the translation of the vehicle, resulting in minimal differences in $t_{rel}$ compared to the baseline. After the addition of local road plane constraints, further enhancements in $t_{rel}$ and $t_{ate}$. There are two reasons: one direct reason is that the system establishes a local road plane to constrain the motion of the vehicle. Consequently, the motion estimation of the vehicle is closely with real physical conditions, thereby reducing vertical drift. The other indirect reason is that during the local road modeling process, observations from previous frames are utilized. This strengthens the correlation between the current frame and previous frames, consequently enhancing the inter-frame scale consistency of the system.

**Table 1.** Experimental evaluation on the KITTI−360 dataset [$t_{ate}$ (m), $t_{rel}$ (%), $r_{rel}$ (° / 100 m)], without loop correction.

| Seq. | Start/Stop Frame | VSLAM | | | VSLAM+EG | | | VSLAM+EG+LRC (RC-SLAM) | | | Stereo ORB-SLAM2 | | | Stereo OV²SLAM | | | Stereo-Inertial ORB-SLAM3 | | |
|---|---|---|---|---|---|---|---|---|---|---|---|---|---|---|---|---|---|---|---|
| | | $t_{ate}$ | $t_{rel}$ | $r_{rel}$ | $t_{ate}$ | $t_{rel}$ | $r_{rel}$ | $t_{ate}$ | $t_{rel}$ | $r_{rel}$ | $t_{ate}$ | $t_{rel}$ | $r_{rel}$ | $t_{ate}$ | $t_{rel}$ | $r_{rel}$ | $t_{ate}$ | $t_{rel}$ | $r_{rel}$ |
| 00 | 1125/4142 | 5.65 | 0.46 | 0.42 | 5.15 | 0.44 | 0.30 | **4.85** | **0.35** | 0.32 | 5.05 | 0.38 | 0.27 | 5.23 | 0.45 | 0.31 | 5.54 | 0.43 | **0.24** |
| 02 | 11,432/12,944 | 4.45 | 0.42 | 0.38 | 4.05 | 0.39 | 0.25 | **2.94** | **0.30** | **0.20** | 3.57 | 0.42 | 0.19 | 4.76 | 0.51 | 0.21 | 4.17 | 0.53 | 0.23 |
| 04 | 6473/9890 | 6.20 | 0.61 | 0.43 | 5.79 | 0.52 | 0.35 | 5.25 | **0.35** | 0.34 | 5.65 | 0.60 | 0.25 | 6.05 | 0.65 | 0.23 | **4.99** | 0.42 | **0.20** |
| 05 | 965/3214 | 4.23 | 0.49 | 0.35 | 4.01 | 0.45 | 0.32 | 3.52 | 0.40 | 0.31 | 3.67 | 0.48 | **0.27** | 4.02 | 0.46 | 0.30 | **3.20** | **0.32** | 0.28 |
| 06 | 611/2484 | 4.26 | 0.53 | 0.33 | 3.99 | 0.45 | 0.3 | **3.58** | **0.38** | 0.26 | 4.19 | 0.49 | **0.23** | 4.65 | 0.52 | 0.31 | 4.32 | 0.54 | 0.24 |
| 07 | 3/2030 | 10.52 | 1.00 | 0.48 | 8.65 | 1.58 | 0.41 | **7.54** | 1.25 | **0.38** | 12.52 | 2.25 | 0.65 | 8.75 | 1.25 | 0.39 | 8.02 | **0.98** | 0.47 |
| 09 | 1847/4711 | 8.95 | 0.70 | 0.45 | 5.95 | 0.65 | 0.29 | **5.25** | 0.46 | **0.26** | 6.25 | 0.62 | 0.29 | 5.05 | **0.45** | 0.29 | 5.98 | 0.48 | 0.33 |

The best results are shown in bold.

Table 1 also demonstrates the evaluation results of the proposed system compared to three open-source systems. The proposed system attained the best results across most sequences. Overall, the proposed system outperformed both ORB-SLAM2 and OV$^2$SLAM, two stereo Visual SLAM systems. Moreover, in terms of $t_{ate}$, it shows advantages compared to the stereo visual system that fuses IMU. Contrasting with ORB-SLAM2 and OV$^2$SLAM, the proposed system shows improvements in $t_{rel}$ in sequences 00, 02, 04, 05, and 06. This suggests that the road constraint can effectively reduce inter-frame drift. Simultaneously, there is a slight elevation in $r_{rel}$. This is because the proposed system utilizes more road feature points closer to the vehicle. This led to more effective constraints for enhancing the estimation of rotation. Compared to the ORB-SLAM3, which fuses stereo vision and IMU, the proposed system placed greater emphasis on using constraints from roads and previous frames. This led to an enhancement in global scale consistency. Therefore, in $t_{ate}$ metric, RC-SLAM with a stereo camera could achieve slightly better performance than the IMU-integrated ORB-SLAM3.

For a more intuitive comparison between the proposed system and three comparative systems aligned with ground truth, Figures 7 and 8 show the estimated trajectories with the ground truth for KITTI-360 dataset sequences 00 and 02. It is evident that the scale of RC-SLAM is closer to the ground truth compared to the other comparative methods. This aligns with the smaller absolute trajectory error achieved by the proposed method, as shown in Table 1. Figure 9 further shows the comparison between RC-SLAM and the comparative methods in y position against ground truth for KITTI-360sequence 020. The proposed system better matches the ground truth in the vertical orientation of the vehicle. This substantiates that road constraints can reduce the vertical drift in vehicle motion.

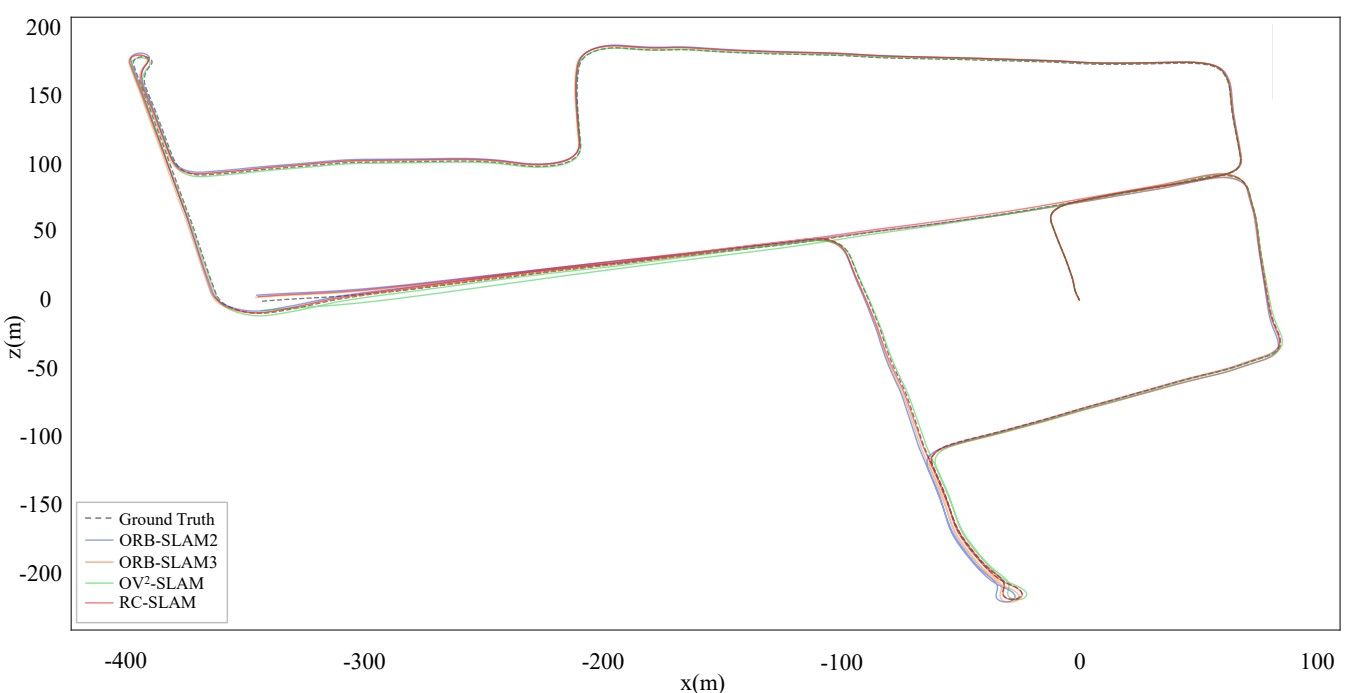

**Figure 7.** Estimated trajectories and ground truth for KITTI−360 sequence 00.

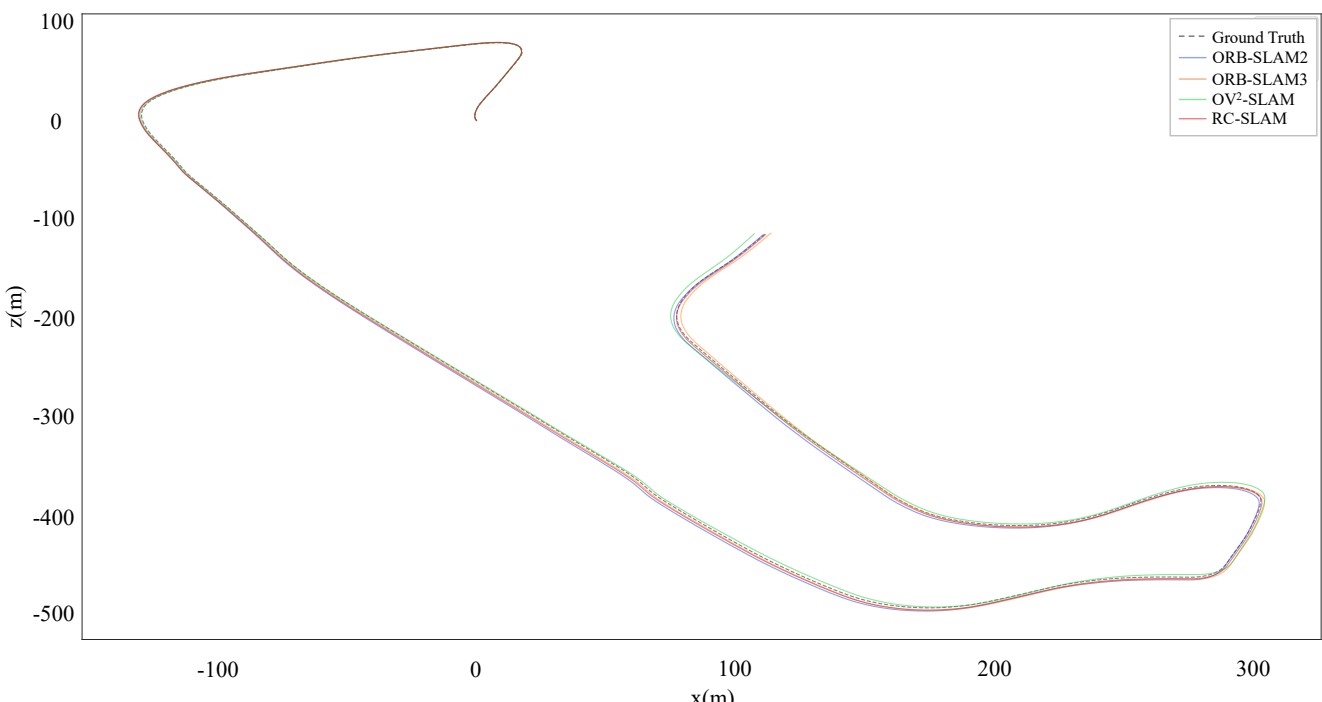

**Figure 8.** Estimated trajectories and ground truth for KITTI−360 sequence 02.

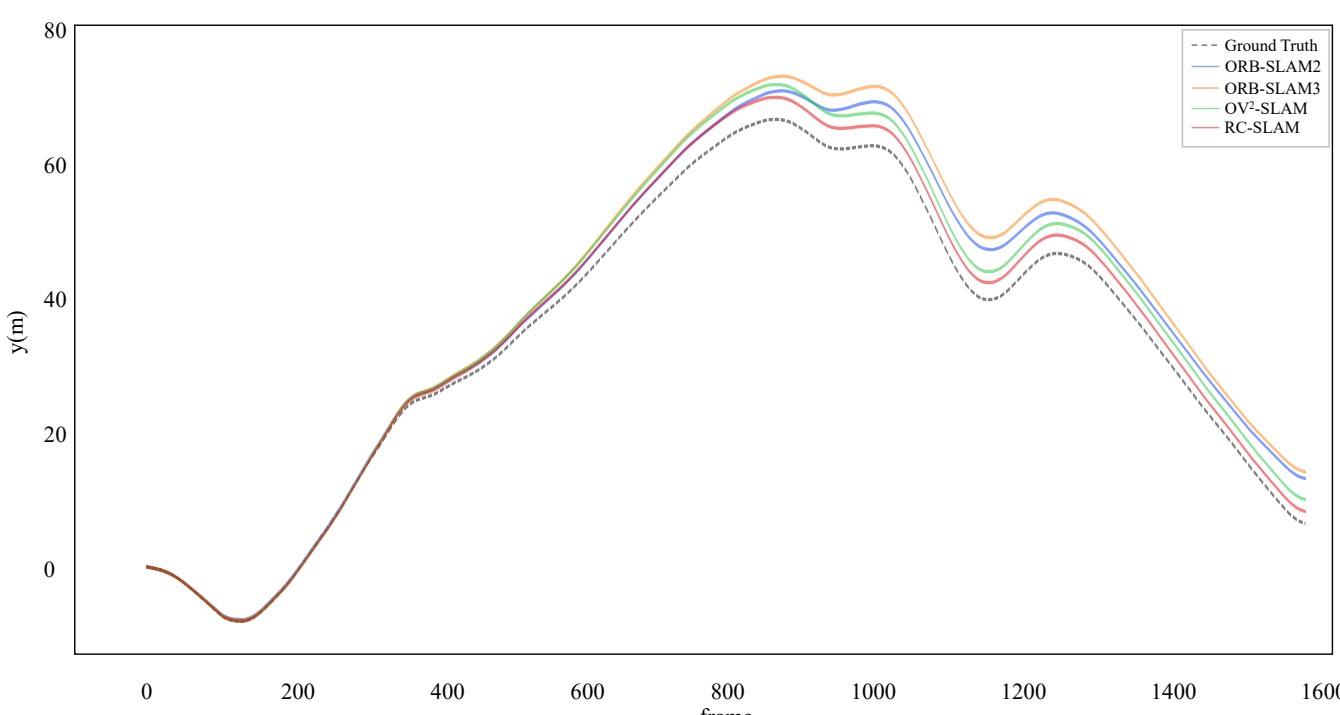

**Figure 9.** Estimated y positions and ground truth for KITTI−360 sequence 02.

## 5.2. KITTI Dataset

The KITTI Odometry dataset comprises 11 urban driving scenes with ground truth, including highways, urban streets, and residential areas. It includes data from stereo cameras (color and grayscale), Lidar, and IMU. In this experiment, rectified stereo color images were used, captured by a stereo camera with a baseline of 0.54 m, resolution of $1392 \times 512$ pixels, and frequency of 10 Hz. The high-precision ground truth of the vehicle generated by an OXTS3003 GPS/IMU unit was employed to evaluate the trajectories of

SLAM systems. On the KITTI dataset, the proposed system and comparative system were tested under two conditions: with loop correction disabled and enabled. As the KITTI dataset contains loop closure scenes in sequences 00, 02, 05, 08, and 09, the performance of each method was assessed with loop correction in these five sequences.

Table 2 presents the results of the proposed method, ORB-SLAM2, OV²SLAM, and ORB-SLAM3 without loop correction. The proposed system gets similar results to the KITTI360 dataset. Across metrics like $t_{rel}$ and $r_{rel}$, the proposed system achieves the best results in most sequences. We attribute this to the proposed local road plane constraint and epipolar constraint for road features, which enhance accuracy in rotation and translation estimations. Compared to ORB-SLAM3, which fuses IMU, in scenes where ORB-SLAM3 initializes smoothly (like sequences 03 and 05), the proposed system exhibits higher $t_{ate}$. However, the proposed system gets similar or even better results in other sequences than ORB-SLAM3. These outcomes suggest that while systems fused with IMU demonstrate increased accuracy, the prolonged or failed IMU initialization affects the entire SLAM system in some scenes. In contrast, the proposed system, independent of other sensors, explicitly expresses the physical constraints between the vehicle and the road, thus enhancing the accuracy and robustness of the system. As illustrated in Figures 10 and 11, the trajectories estimated by the proposed system closely align with the ground truth. In sequence 03, the proposed system gets the lower error in the y position compared to the three comparative systems.

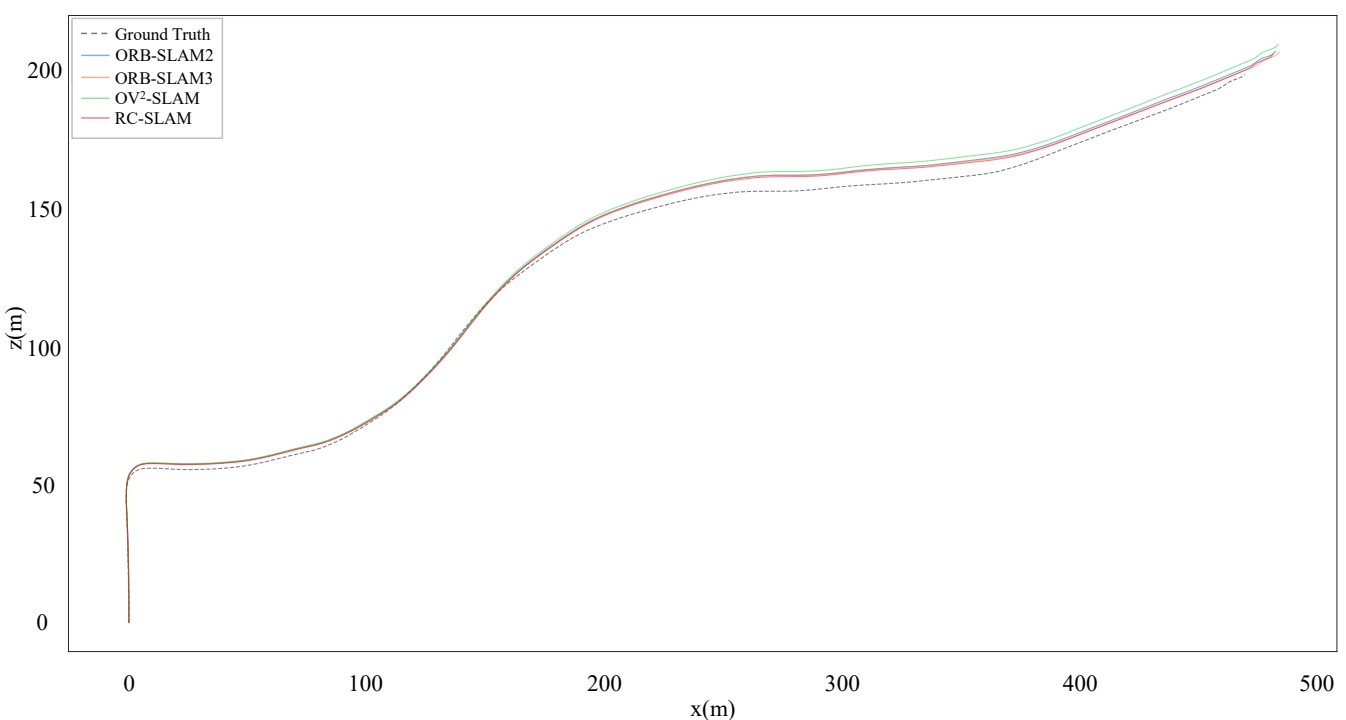

**Figure 10.** Estimated trajectories and ground truth for KITTI sequence 03.

Table 3 shows the experimental results of RC-SLAM, ORB-SLAM2, OV²SLAM, and ORB-SLAM3 on sequences 00, 02, 05, 06, 08, and 09 of the KITTI dataset with loop correction. All four systems detected and underwent loop correction in these six sequences. Loop correction effectively mitigates accumulated drift in trajectories, resulting in more consistent and accurate overall trajectories. Consequently, there is a notable improvement in $t_{ate}$ for all four systems. Although the proposed system gets the best result of $t_{ate}$ only in sequence 08, it consistently demonstrates second-best results in the other five sequences. This demonstrates the effectiveness of using local plane features to represent road characteristics, enhancing the proposed system during global Bundle Adjustment (BA). Combined with loop closure's ability to constrain all frames within the loop, the global consistency of

the system is further improved. Figure 12 presents a comparison between RC-SLAM and the comparative systems in terms of estimated trajectories against ground truth after enabling loop closure, demonstrating higher consistency between the proposed method and the ground truth across the entire trajectory. This is due to the fact that within shorter frame sequences, local road plane features also contribute to inter-frame constraints, enhancing inter-frame scale consistency. When combined with loop correction, the proposed system gets better consistency. Figure 12 presents a comparison between estimated trajectories and ground truth with loop correction, demonstrating higher consistency between the proposed method and the ground truth.

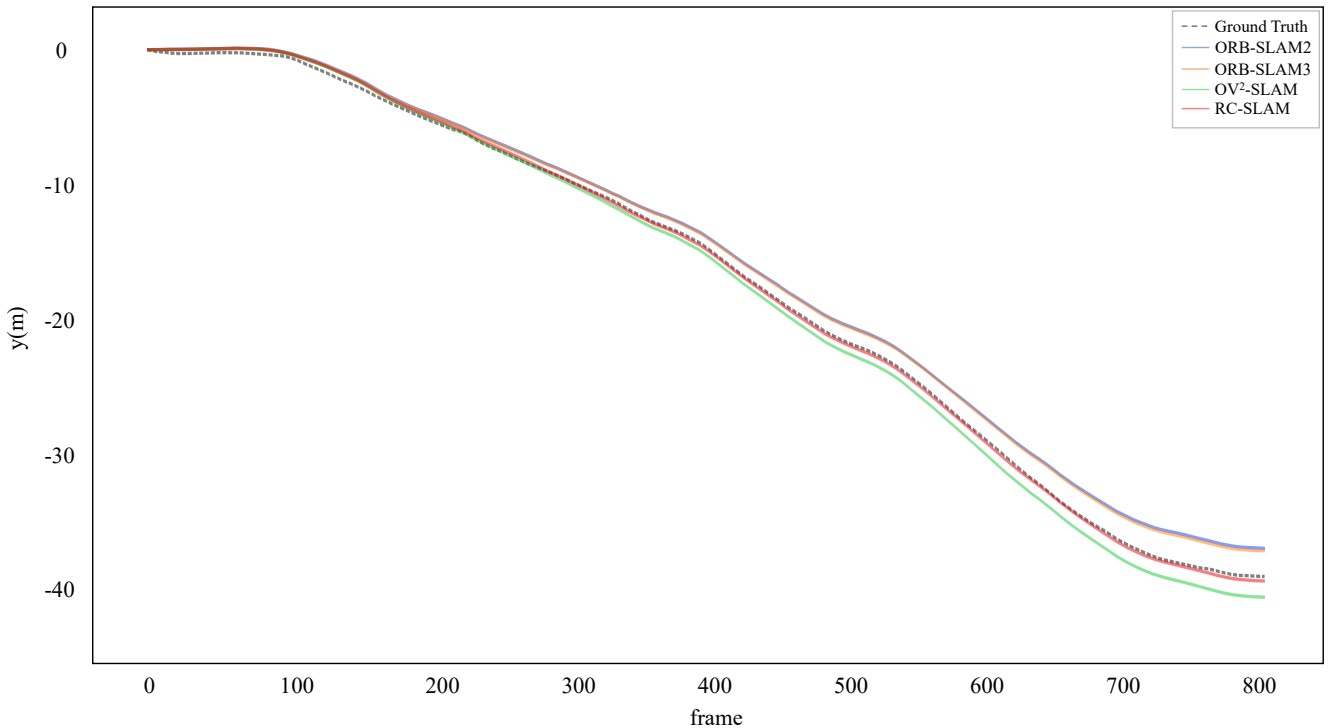

**Figure 11.** Estimated y positions and ground truth for KITTI sequence 03.

**Table 2.** Experimental evaluation on the KITTI dataset $[t_{ate}(\mathrm{m}), t_{rel}(\%), r_{rel}(°/100\,\mathrm{m})]$, without loop correction.

| Seq. | RC-SLAM | | | Stereo ORB-SLAM2 | | | Stereo OV²SLAM | | | Stereo-Inertial ORB-SLAM3 | | |
|---|---|---|---|---|---|---|---|---|---|---|---|---|
| | $t_{ate}$ | $t_{rel}$ | $r_{rel}$ | $t_{ate}$ | $t_{rel}$ | $r_{rel}$ | $t_{ate}$ | $t_{rel}$ | $r_{rel}$ | $t_{ate}$ | $t_{rel}$ | $r_{rel}$ |
| 00 | **4.10** | **0.89** | **0.52** | 5.59 | 0.92 | 0.60 | 7.52 | 0.99 | 0.59 | 6.36 | 0.92 | 0.52 |
| 01 | 8.74 | 0.87 | **0.27** | 10.95 | 1.45 | 0.28 | 8.24 | **0.85** | 0.29 | **7.62** | 0.87 | 0.29 |
| 02 | **8.03** | 1.03 | **0.43** | 8.77 | **0.99** | 0.44 | 9.53 | 1.15 | 0.44 | 8.34 | 1.03 | 0.44 |
| 03 | 3.52 | **0.67** | 0.34 | 4.78 | 0.73 | 0.33 | 4.59 | 0.98 | **0.25** | **3.24** | 0.76 | 0.30 |
| 04 | **0.99** | **0.31** | **0.10** | **0.99** | 0.33 | 0.11 | 1.32 | 0.34 | 0.16 | 1.61 | 0.37 | 0.13 |
| 05 | 4.02 | **0.46** | 0.42 | 4.52 | 0.48 | 0.42 | 4.60 | 0.52 | 0.40 | **3.02** | 0.51 | **0.38** |
| 06 | **2.80** | 0.57 | **0.29** | 3.14 | **0.55** | 0.39 | 4.74 | 0.66 | 0.31 | 3.32 | 0.57 | 0.29 |
| 07 | **1.04** | 0.47 | **0.24** | 1.33 | 0.44 | 0.26 | 2.60 | 0.86 | 0.50 | 1.38 | **0.43** | 0.25 |
| 08 | 5.22 | **1.12** | **0.47** | 5.32 | 1.18 | 0.49 | 7.00 | 1.31 | 0.47 | **5.06** | 1.23 | 0.49 |
| 09 | **3.85** | 0.69 | 0.37 | 4.96 | 0.72 | 0.39 | 4.08 | 0.79 | **0.30** | 4.48 | **0.68** | 0.38 |
| 10 | 1.98 | **0.82** | **0.34** | 2.08 | 0.84 | 0.43 | **1.94** | 0.83 | 0.36 | 2.30 | 0.83 | 0.45 |

The best results are shown in bold.

**Table 3.** Experimental evaluation on the KITTI dataset $[t_{ate}(\text{m}), t_{rel}(\%), r_{rel}(^\circ/100\text{ m})]$, with loop correction.

| Seq. | RC-SLAM | | | Stereo ORB-SLAM2 | | | Stereo OV$^2$SLAM | | | Stereo-Inertial ORB-SLAM3 | | |
|---|---|---|---|---|---|---|---|---|---|---|---|---|
| | $t_{ate}$ | $t_{rel}$ | $r_{rel}$ | $t_{ate}$ | $t_{rel}$ | $r_{rel}$ | $t_{ate}$ | $t_{rel}$ | $r_{rel}$ | $t_{ate}$ | $t_{rel}$ | $r_{rel}$ |
| 00 | 1.8 | **0.9** | **0.45** | **1.75** | 0.91 | 0.6 | 2.67 | 1.28 | 0.56 | 1.8 | 0.9 | 0.52 |
| 02 | 5.77 | **0.97** | **0.39** | **5.46** | 0.98 | 0.4 | 8.75 | 1.5 | 0.5 | 5.6 | 0.99 | 0.42 |
| 05 | 0.73 | **0.42** | 0.34 | 0.69 | **0.42** | **0.3** | 2.12 | 1.17 | 0.48 | **0.64** | 0.43 | **0.3** |
| 06 | 1.29 | 0.57 | 0.54 | 1.5 | **0.53** | 0.34 | **1.26** | 0.97 | 0.43 | 1.39 | 0.54 | **0.28** |
| 08 | **2.9** | 1.19 | 0.51 | 3.1 | **1.17** | 0.51 | 3.66 | 1.31 | **0.47** | 3.32 | 1.2 | 0.5 |
| 09 | 2.91 | 0.71 | **0.3** | 2.61 | 0.68 | 0.34 | 3.29 | 0.69 | **0.3** | 2.77 | **0.63** | 0.31 |

The best results are shown in bold.

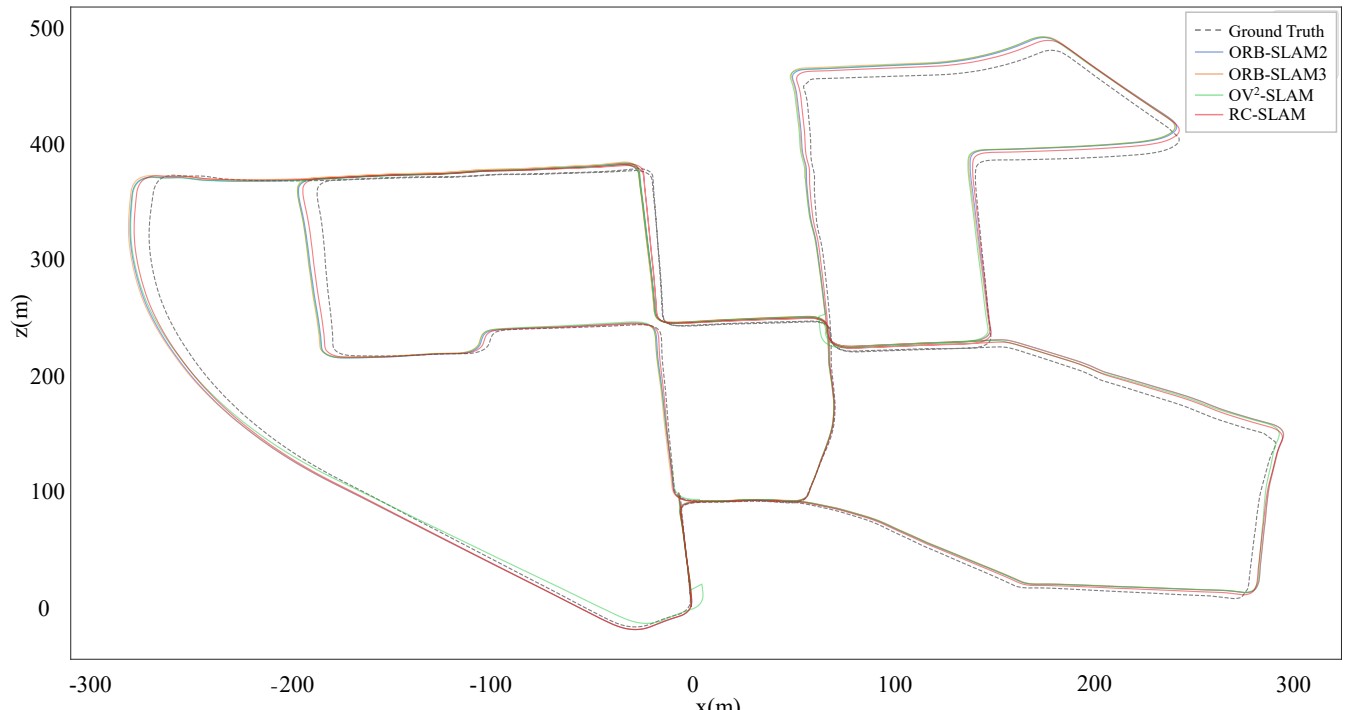

**Figure 12.** Estimated trajectories and ground truth for KITTI sequence 00.

### 5.3. Real-World Experiments

Data within real-world scenes is gathered by a data collection vehicle. The four sequences were all captured within the campus. Among these, Sequence 01 includes a loop closure scene, while the other three lack it. The data collection vehicle is equipped with a stereo color camera having a baseline of 0.2 m, resolution of $1280 \times 720$ and a frame rate of 30 Hz. It is also equipped with an Xsens MTI-300 IMU operating at a frame rate of 200 Hz, a LiDAR with a frequency of 10 Hz, and a Bynav GNSS/IMU Unit. Additionally, wheel speed and steering angle are acquired from the CAN bus of the vehicle. Data from all sensors are recorded using a Data Logger. The extrinsics among different sensors and the intrinsic of the stereo camera were calibrated before the experiment. The data collection vehicle and various sensors are shown in Figure 13. In this experiment, we utilized images from the stereo color camera and the output as ground truth from the Bynav GNSS/IMU Unit which underwent coordinate transformation, time synchronization, and other processing. It is noteworthy that unlike the stereo cameras employed in KITTI and KITTI-360, the baseline of the camera on the data collection vehicle is only 0.2 m, enabling the real-world data to reflect the performance of various systems with a smaller baseline camera.

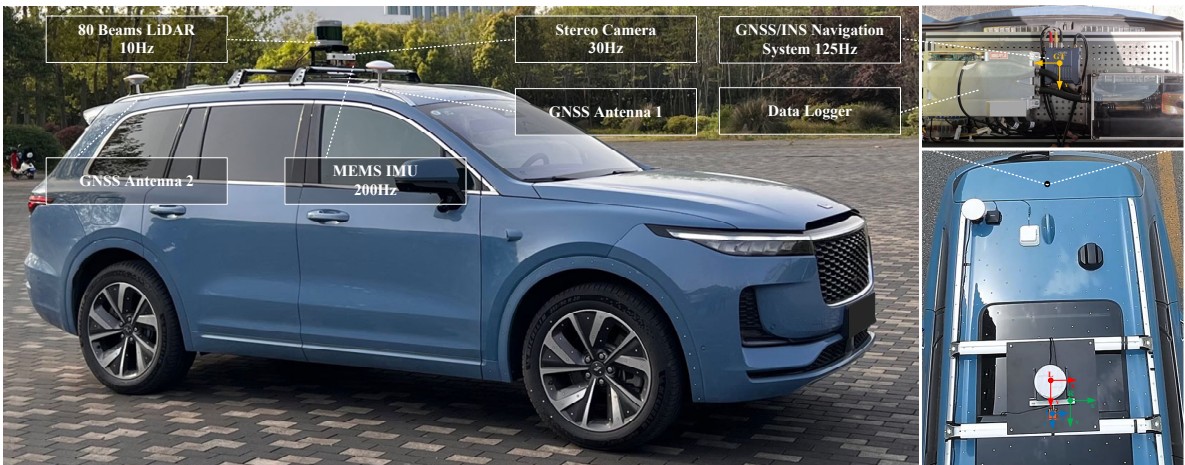

**Figure 13.** The data collection vehicle and equipped sensors.

Table 4 shows the experimental results of RC-SLAM, ORB-SLAM2, OV2-SLAM, and ORB-SLAM3 within the Real-world dataset captured in the campus environment. All systems detected loop closures and performed loop corrections in Sequence 01, estimated trajectories and ground truth were shown in Figure 14. The proposed system exhibited the minimum $t_{ate}$ in datasets 00, 01, and 02, and achieved a near-optimal result in dataset 03. As shown in Figure 15, this indicates that the proposed system can achieve better global consistency even with a small baseline camera. This performance is still dependent on assistance from the local road plane constraint. However, due to the reduced camera baseline, the number of nearby feature points in the front-end significantly diminishes, unavoidably leading to decreased accuracy in rotational estimation. Nevertheless, within RC-SLAM, employing 2D features for matching ground feature points allows the acquisition of more nearby feature points. When combined with epipolar constraints, this yields more accurate rotational estimation. Consequently, in Sequences 01 and 03, the proposed method also achieved the best outcomes in terms of relative average rotational error, while obtaining a second-best result in Sequence 02.

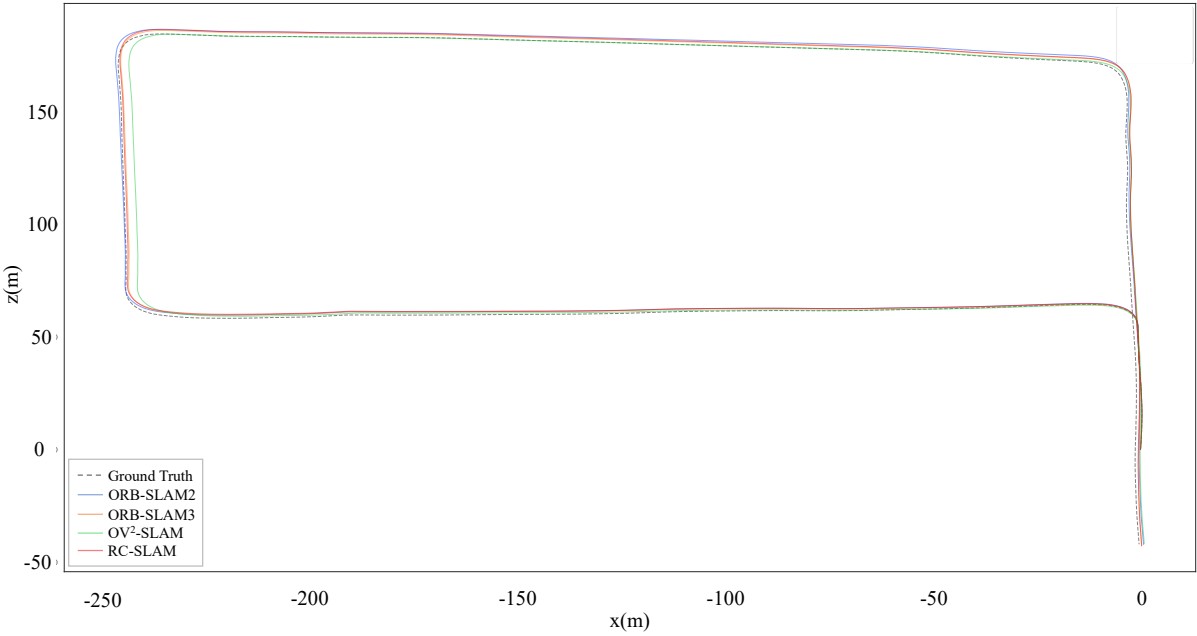

**Figure 14.** Estimated trajectories and ground truth for sequence 01.

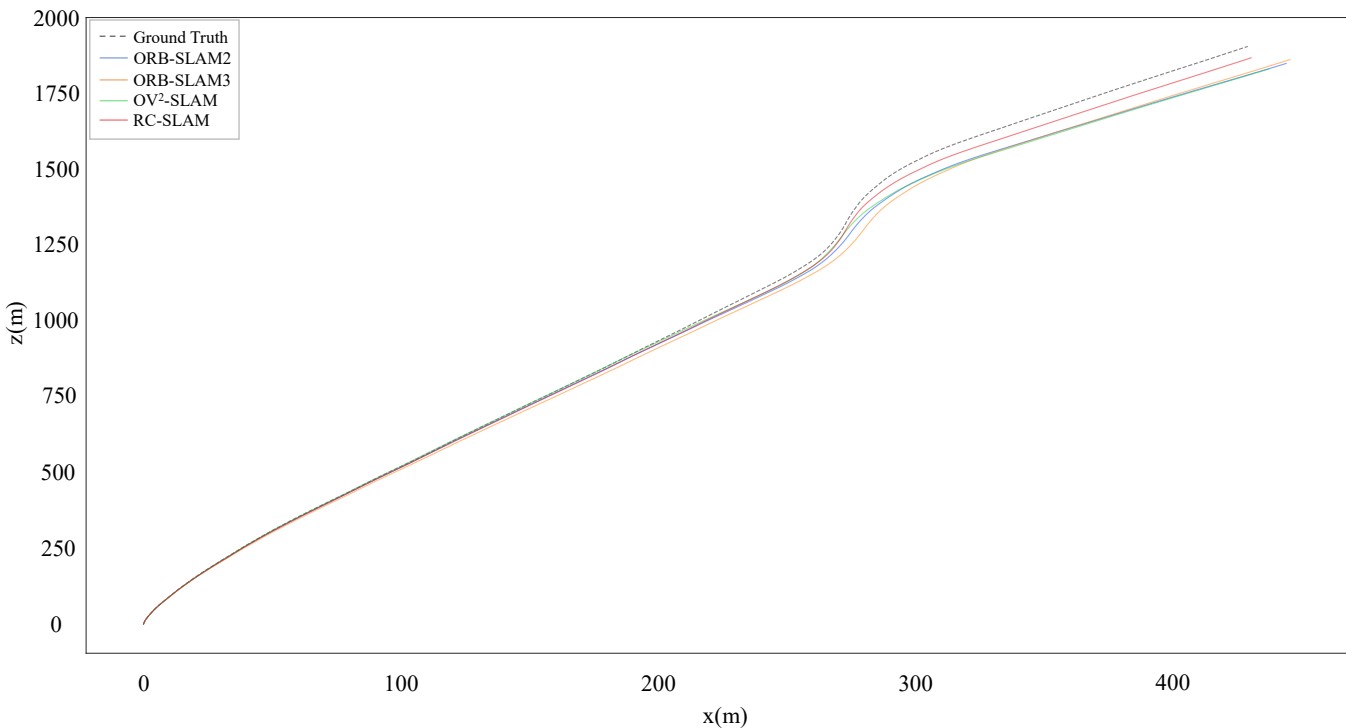

**Figure 15.** Estimated trajectories and ground truth for sequence 02.

**Table 4.** Experimental evaluation on real-world dataset $[t_{ate}(\mathrm{m}), t_{rel}(\%), r_{rel}(°/100\,\mathrm{m})]$, with loop correction.

| Seq. | RC-SLAM | | | Stereo ORB-SLAM2 | | | Stereo OV$^2$-SLAM | | | Stereo-Inertial ORB-SLAM3 | | |
|---|---|---|---|---|---|---|---|---|---|---|---|---|
| | $t_{ate}$ | $t_{rel}$ | $r_{rel}$ | $t_{ate}$ | $t_{rel}$ | $r_{rel}$ | $t_{ate}$ | $t_{rel}$ | $r_{rel}$ | $t_{ate}$ | $t_{rel}$ | $r_{rel}$ |
| 00 | **3.19** | **0.82** | 0.26 | 4.48 | 0.95 | 0.29 | 3.72 | 1.01 | 0.26 | 4.96 | 1.21 | **0.16** |
| 01 | **0.52** | 0.47 | **0.20** | 0.53 | **0.45** | 0.26 | 0.60 | 0.51 | 0.21 | 1.00 | 0.63 | 0.26 |
| 02 | **2.73** | **0.78** | 0.41 | 2.87 | 1.01 | **0.40** | 2.80 | 1.35 | 0.42 | 8.75 | 2.51 | 0.48 |
| 03 | 3.80 | 0.86 | **0.23** | 4.04 | 0.86 | 0.28 | 3.92 | **0.79** | 0.24 | **3.62** | 1.04 | 0.25 |

The best results are shown in bold.

## 6. Conclusions

In this work, a stereo Visual SLAM system with road constraints based on graph optimization was proposed for intelligent vehicles. Firstly, the proposed system fully utilizes the matched road feature point between keyframes to construct epipolar constraints, which can avoid the impact of depth uncertainty of road feature points on the system and thereby achieve more accurate rotation estimation. Secondly, the system employs observations of the local road corresponding to the current keyframe from previous keyframes to estimate parameters of the local road plane and establishes constraints on the vehicle based on this plane. Lastly, the system obtains precise vehicle poses and global maps by utilizing nonlinear optimization to jointly optimize vehicle trajectories, LPRs, and map points. The ablation experiments demonstrate that the two road constraints in the system, focusing on epipolar constraints and local road constraints, effectively reduce errors arising from the xis DoF motion assumption of the vehicle. By comparing the proposed system with state-of-the-art Visual SLAM and Visual-inertial SLAM on the KITTI-360 dataset and KITTI dataset, the proposed system achieved more accurate trajectories of vehicles without the addition of extra sensors. Finally, further validation of the proposed system was demonstrated in real-world experiments. In future work, the system needs to be tested in more real-world road scenes. Moreover, the numerous dynamic objects on the road affect the localization

and mapping of the system during experiments. To address this problem, dynamic SLAM is a worthwhile research direction.

**Author Contributions:** In this article, the authors' contributions are shown below: conceptualization, Y.Z. and H.A.; methodology, Y.Z. and H.A.; formal analysis, H.W. and M.W.; supervision, K.L.; writing-original draft, H.A.; writing—review and editing, R.X. All authors have read and agreed to the published version of the manuscript.

**Funding:** This work was supported by the Perspective Study Funding of Nanchang Automotive Institute of Intelligence and New Energy, Tongji University (grant number: TPD-TC202211-07).

**Institutional Review Board Statement:** Not applicable.

**Informed Consent Statement:** Not applicable.

**Data Availability Statement:** Publicly available datasets were analyzed in this study. These data can be found here: https://www.cvlibs.net/datasets/kitti/eval_odometry.php (accessed on 2012) and https://www.cvlibs.net/datasets/kitti-360/index.php (accessed on 2022).

**Acknowledgments:** We appreciate the critical and constructive comments and suggestions from the reviewers that helped improve the quality of this manuscript. We also would like to offer our sincere thanks to those who participated in the data processing and provided constructive comments for this study.

**Conflicts of Interest:** The authors declare no conflicts of interest.

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
