# Peer review of "RC-SLAM: Road Constrained Stereo Visual SLAM System Based on Graph Optimization"

_sensors, doi:10.3390/s24020536_

Round 1

Reviewer 1 Report

Comments and Suggestions for Authors

1. The paper could benefit from providing more details on the proposed method for approximating local roads as discrete planes and extracting parameters of local road planes (LRPs) using homography.

2. It would be helpful to include a more thorough explanation of the epipolar constraints applied to estimate rotation and how they minimize the distance between road feature points and epipolar lines.

3. The paper should provide more information on the nonlinear optimization model based on graph optimization and how it jointly optimizes the poses of vehicle trajectories, LPRs, and map points.

4. To show the importance of the research topic, some related works can be reviewed. For example, "Self-supervised graph completion for incomplete multi-view clustering"; "Centric graph regularized log-norm sparse non-negative matrix factorization for multi-view clustering".

5. The paper should provide more details on the tightly coupled graph optimization framework and how explicit constraints between the vehicle and local road planes are established.

6. It would be beneficial to include a more detailed explanation of the transformation matrix T C k W and how it is used to convert 3D landmarks from the world frame to the camera frame.

7. The authors should consider providing more information on the datasets used in the experiments and how they were selected.

Comments on the Quality of English Language

Minor editing of English language required

Reviewer 2 Report

Comments and Suggestions for Authors

The problems with the paper are summarized as follows:

1.Figure 2 should belong to the fourth part, not above it.

2.The workload of this article is acceptable, but the innovation is slightly insufficient. It is recommended that the author highlight the contribution of the proposed method and the different support from previous methods when describing the work in this article, in order to highlight the importance of the work in this article.

3.“ Consequently, in Sequences 01 and 03, the proposed method also  achieved the best outcomes in terms of relative average rotational error, while obtaining a second-best result in Sequence 02.” What is the reason why the method proposed in this paper cannot achieve the best results on the second sequence? Suggest the author to further improve the algorithm in this article, improve the accuracy of the method, and demonstrate the universality and absolute advantages of the algorithm.                                                                    4.“To minimize the impact of ran-domness in each system, each system was consecutively run five times on every sequence of the dataset.” Five times is not enough,and suggest the author do experiments more times. Comments on the Quality of English Language

Minor editing of English language required.
